# Loliolide Presents Antiapoptosis and Antiscratching Effects in Human Keratinocytes

**DOI:** 10.3390/ijms20030651

**Published:** 2019-02-02

**Authors:** Sang Hee Park, Dong Sam Kim, Sunggyu Kim, Laura Rojas Lorz, Eunju Choi, Hye Yeon Lim, Mohammad Amjad Hossain, SeokGu Jang, Young Im Choi, Kyung Ja Park, Keejung Yoon, Jong-Hoon Kim, Jae Youl Cho

**Affiliations:** 1Department of Biocosmetics, Sungkyunkwan University, Suwon 16419, Korea; 84701@naver.com (S.H.P.); sukim590@skku.edu (S.K.); gosl177@naver.com (H.Y.L.); 2Samcheok Prasiola Japonica Research Center, Samcheok City Hall, Samcheok 25914, Korea; prasiolra@korea.kr (D.S.K.); jangsg69@korea.kr (S.J.); Prism@korea.kr (Y.I.C.); kyu5132@korea.kr (K.J.P.); 3Research and Business Foundation, Sungkyunkwan University, Suwon 16419, Korea; 4Department of Integrative Biotechnology, Sungkyunkwan University, Suwon 16419, Korea; laurisrl@gmail.com (L.R.L.); cej223@naver.com (E.C.); keejung@skku.edu (K.Y.); 5Department of Veterinary Physiology, College of Medicine, Chonbuk National University, Iksan 54596, Korea; mamjadh2@gmail.com

**Keywords:** antiapoptosis, antiscratching effect, antiwrinkling activity, antioxidant activity

## Abstract

Loliolide is a monoterpenoid hydroxylactone present in freshwater algae that has anti-inflammatory and antiaging activity; however, its effects on ultraviolet-damaged skin have yet to be elucidated. This study investigated the antiapoptosis and wound-healing effects of loliolide using HaCaT cells (a human keratinocyte cell line). Loliolide inhibited the expression of reactive oxygen species (ROS) induced by ultraviolet radiation as well as wrinkle formation-related matrix metalloproteinase genes and increased the expression of the damage repair-related gene SIRT1. The apoptosis signaling pathway was confirmed by Western blot analysis, which showed that loliolide was able to reduce the expression of caspases 3, 8, and 9, which are related to ROS-induced apoptosis. In addition, Western blotting, reverse-transcription polymerase chain reaction (PCR), and real-time PCR analyses showed that loliolide enhanced the expression of the epidermal growth factor receptor signaling pathway (PI3K, AKT) and migration factors, such as K6, K16, and K17; keratinocyte growth factor; and inflammatory cytokines, such as interleukin (IL)-1, IL-17, and IL-22 expressed during the cellular scratching process, suggesting a putative wound-healing ability. Because of the antiapoptosis and antiscratching effects on skin of both loliolide and loliolide-rich *Prasiola japonica* ethanol extract, we consider the former to be an important compound used in the cosmeceutical industry.

## 1. Introduction

Skin is a self-renewing organ that defends the body against external threats such as heat, infections, mechanical or chemical insults, and ultraviolet (UV) radiation [1,2]. UV radiation, which causes cell senescence and death, has been reported as an important cause of skin aging, wrinkling, and pigmentation [3]. Skin exposure to UV can induce erythema, DNA damage, formation of reactive oxygen species (ROS), and skin inflammation and wrinkling, among other outcomes [4,5]. ROS generation has been proven to be related to the skin’s collagen degradation through the activation of matrix metalloproteinases (MMPs) [6], which leads to both skin sagging and wrinkling. UV exposure can induce the expression of the cyclooxygenase-2 (COX-2) gene, which causes skin inflammation, as well as to the downregulation of Sirt-1, a gene involved in UV-induced DNA damage repair, cell survival, cell metabolism, and senescence [7]. In addition, UVB-generated ROS can trigger cell apoptosis [8,9], a process of programmed cell death, which involves a series of morphological changes, including cell detachment, cell shrinkage, chromatin condensation, and DNA fragmentation [10]. The apoptosis signaling pathway mechanism is known to be controlled by proteolytic enzymes such as caspases 3, 8, and 9 [11]. Since both skin wrinkling and apoptosis can be induced by ROS, one approach to maintaining healthy skin and preventing apoptosis-induced skin aging is the use of antioxidants; therefore, compounds with high antioxidant activity are in high demand in the cosmeceutical industry [12,13].

Since the skin is the outer barrier of our body, another of its functions is to protect our body from potential injury and, when injury occurs, to promote quick restoration to prevent potential infections [14]. Therefore, compounds that can accelerate skin repair and regeneration are highly desirable. Cutaneous wound healing is an essential physiological process during which the skin attempts to mitigate a lesion induced by a local aggression, which is achieved by the deposition of collagen, cell proliferation, and posterior differentiation [15]. One major wound-healing process is triggered by the activation of the epidermal growth factor receptor (EGFR) pathway, which includes the phosphorylation of the PI3K, AKT, and ERK proteins and which has been shown to control diverse cellular activities, including cell survival, growth, proliferation, metabolism, and migration [16]. 

Wound healing begins with the induction of an inflammatory response due to bacterial colonization and barrier disruption, which stimulate the immediate production of interleukin (IL)-1, tumor necrosis factor alpha (TNF-α), and Th17 cell-produced cytokines [17]. Th17 cell-produced cytokines such as IL-17 and IL-22 [18] can impact wound healing positively by clearing pathogens and modulating mucosal surfaces and epithelial cells [19], although these cytokines sometimes function negatively in the wound healing process by inducing cellular damages. The inflammatory phase is followed by a tissue formation phase, which includes the activation of K6 and the K6 dimerization partner K16 to achieve epithelialization, proliferation, and migration of the follicular keratinocytes. The keratinocyte growth factor (KGF) simultaneously becomes strongly activated during this phase and induces keratinocyte proliferation and migration for healing the wound [20,21].

Loliolide (Figure 1) is an active ingredient in green algae that has been shown to have antioxidant, cellular protection [22,23] anticancer [24], and cell senescence inhibition [24] activities. In our previous study, we also showed that loliolide demonstrates important antioxidant and antimelanogenic activity, suggesting its potential as a treatment for skin diseases [25]; however, research related to its effects on UV-damaged skin is still lacking [26]. *Prasiola japonica* ethanol extract (Pj–EE) is a loliolide-rich source that has also been shown to possess antioxidant activity [27]. Therefore, the main objective of the present study was to investigate the effects of loliolide on keratinocytes with UV-induced damage. Because of its antiaging, antioxidant, anti-inflammatory, antiapoptotic, and antiscratching effects, loliolide has great potential as a putative wrinkle-improving and wound-healing compound in the cosmeceutical industry.

## 2. Results

### 2.1. Antiapoptotic and Antioxidant Effects of Loliolide in HaCaT Cells

According to the 3-(4,5-dimethylthiazole-2-yl)-2,5-diphenyltetrazolium bromide (MTT) assay, HaCaT cell viability was not affected even up to a loliolide concentration of 100 µM under normal cell culture conditions (Figure 2a). By contrast, the viability of HaCaT cells was significantly decreased under a UVB irradiation (30 mJ/cm^2^) condition (Figure 2b). However, pretreated loliolide (100 µM) clearly recovered the down-regulated level of cell viability of UV-irradiated HaCaT cells, implying a putative photoprotective effect against cell death caused by UVB-induced oxidative stress (Figure 2b). Next, we identified the level of expression of ROS through H2-DCFDA staining in HaCaT cells, according to which loliolide could decrease the increase of ROS caused by UVB concentration (Figure 2c). In addition, loliolide was found to inhibit DNA damage as well as ROS inhibition according to DAPI staining assay (Figure 2d). To confirm the inhibitory effects of loliolide on cell death in HaCaT cells, propidium iodide (PI)–annexin V staining and FACS were performed. While UVB treatment induced cell death in HaCaT, pretreatment with loliolide decreased cell death in a dose-dependent manner. (Figure 2e). Furthermore, to confirm the antiapoptosis mechanism of loliolide, we treated HaCaT cells with UVB to induce oxidative stress. When the UVB-treated cells were pretreated with loliolide, the expression of caspases, an important apoptosis gene, was decreased in a dose-dependent manner (Figure 2f).

### 2.2. Effect of Loliolide on MMP Expression in Hacat Cells

We examined the expression of oxidative stress-induced MMPs and Sirt-1 in HaCaT cells to determine loliolide’s involvement in senescence and cell death processes. Loliolide was found to reduce the expression of MMP-1 (Figure 3a), MMP-9 (Figure 3a), MMP-2 (Figure 3b), and MMP-3 (Figure 3c) as well as restoring Sirt-1 expression (Figure 3a) in UVB-irradiated HaCaT cells. Statistical significance was evaluated using the Kruskal–Wallis/Mann–Whitney U test. ## *p* < 0.05 compared with the normal group and ** *p* < 0.01 compared with the control group.

### 2.3. Effect of Loliolide on Cellular Scratching and Cell Migration

To determine the wound-healing effects of loliolide in HaCaT cells, cell migration was measured. In this respect, we confirmed that loliolide (100 µM) showed an increase in cell migration and wound scratch closure in HaCaT cells (Figure 4a,b). To determine cell proliferation, an MTT assay was conducted in loliolide-pretreated HaCaT cells, which resulted in a significantly increased cell proliferation rate, therefore indicating a positive effect in the wound-healing process (Figure 4c). Next, we wanted to examine the effect of loliolide on the expression of genes encoding migration factors (K6, K16, and K17) as well as cell damage and inflammation (TNF-α, MMP-1, and interleukins) in HaCaT cells (Figure 4d–g). Notably, while loliolide increased the expression of migration factors such as K6, K16, K17 (Figure 4d), and KGF (Figure 4h), it showed a significant downregulation in the expression of inflammation-related genes MMP-1 and TNF-α (Figure 4d), while IL-1, IL-17, and IL-22 genes were dose-dependently increased (Figure 4e–g). We examined the expression of proteins in the EGFR signaling pathway in previously scratched HaCaT cells, revealing an increase in the activation of PI3K and AKT proteins (Figure 4i).

### 2.4. Effect of Pj–EE on Cellular Scratching and Cell Migration

We examined the antiscratching effects of Pj–EE, a freshwater green algae whose main compound is known to be loliolide, in keratinocytes (HaCaT cells). Firstly, cell viability in HaCaT cells was measured during Pj–EE-treated condition. According to MTT assay, the viability of the cells was not decreased by Pj–EE until concentrations of up to 400 µg/mL (Figure 5a). To measure the wound-healing effects of Pj–EE in HaCaT cells, the repairing level of cellular scratching progress was measured using the cell migration assay. According to this, Pj–EE (200 µg/mL) showed an increase in cell migration in HaCaT cells (Figure 5b), suggesting a positive wound-healing effect (Figure 5b,c). In addition, MTT-cell-proliferation assay in HaCaT cells showed that Pj–EE significantly increased the cell proliferation rate in HaCaT cells (Figure 5d). Pj–EE additionally increased the expression of migration factors such as K6, K16, K17 (Figure 5e), and KGF (Figure 5i). On the other hand, it showed a remarkable decrease in the expression of the inflammation-related genes, MMP-1 and TNF-α (Figure 5e), while there was an important increase in IL-1, IL-17, and IL-22 in a dose-dependent manner (Figure 5f–h). Next, we examined the expression of proteins in the EGFR signaling pathway in previously scratched and stressed HaCaT cells, showing an increase in the activation of PI3K and AKT (Figure 5j).

## 3. Discussion

As the outmost layer of the human body, our skin is constantly exposed to UV radiation. UVB radiation (280–320 nm), one of the most damaging types of solar UV emissions, can affect various skin structures, causing wrinkling, roughness, and premature aging [28,29]. Constant UVB exposure is known to cause ROS generation, leading not only to DNA damage, but also skin inflammation and cell apoptosis [2]. ROS-induced cell apoptosis becomes necessary when UVB-exposed cells with irreparable DNA damage are high in number [8] and the organism needs to control their survival [30]. For this reason, research with photoprotective compounds that can confer protection against ROS-induced cell death has become important today. As we showed in our previous study, loliolide exhibited a strong antioxidant activity [25]; since antioxidant compounds often exhibit photoprotective activity as well, in this research, we wanted to determine whether loliolide also had a photoprotective effect. Loliolide, apart from showing no cytotoxic effects in concentrations of up to 100 µg/mL (Figure 2a), showed promising photoprotective activity, since it increased cell survival when subjected to UVB radiation (Figure 2b).

In addition to presenting a good antiapoptotic effect, loliolide also dose-dependently reduced not only oxidative stress (Figure 2d), but also DNA damage (Figure 2e), cell death (Figure 2f), and apoptosis-related caspase activation (Figure 2f) exhibited under UV irradiation conditions. In addition to inducing apoptosis, UVB radiation is also involved in the photoaging and wrinkle formation processes [25], which are activated by inflammatory genes such as TNF-α, and the matrix metalloproteinase family, known for both inhibiting procollagen expression and enhancing collagen degradation [31]. Since loliolide could reduce not only the expression of inflammatory genes TNF-α (Figure 4d) but also the expression of the collagen degradation-related genes MMP-1, MMP-9 (Figure 3a), MMP-2, and MMP-3 (Figure 3b,c), while also restoring Sirt-1 expression (Figure 3a), we can conclude that loliolide has exceptional anti-inflammatory, photoprotective, antiaging, and antiwrinkling activities that make it valuable for the cosmeceutical industry. The same pattern was observed with the loliolide-enriched compound Pj–EE, which presented positive anti-inflammatory activity by downregulating inflammatory genes such as MMP-1 and TNF-α (Figure 5e), confirming loliolide’s potential for biocosmetics use. 

Wound healing is an essential physiological process that involves responses such as inflammation, blood clotting, cellular proliferation, and extracellular matrix (ECM) remodeling, with the final aim of preventing infection and dehydration in the body [14]. The longer it takes for a wound to heal, the higher the risk is for the body to succumb to infection or dehydration; therefore, there is a need for materials that can reduce these risks [15]. According to previous research, it was reported that wound-healing-enhancing compounds usually increase cell proliferation and migration rates as well as enhance the EGFR-mediated signaling pathway, including the PI3K, AKT, and ERK proteins [16,32]. Loliolide was found to have an important wound-healing activity, showing an increase not only in the migration rate under cellular scratching conditions (Figure 4a,b) but also in the cell proliferation rate (Figure 4c) of keratinocytes after being subjected to wound stress. In addition, loliolide could upregulate the expression of migration-related factors such as K6, K16, K17 (Figure 4d), and KGF (Figure 4g) as well as wound healing-promoting cytokines such as IL-1, IL-17, and IL-22 (Figure 4e,f,h) in a dose-dependent manner. This is all in concordance with the EGFR signaling pathway results, which showed an increase in the activation of PI3K and AKT (Figure 4i). The loliolide-rich compound Pj–EE also showed not only low cytotoxicity in concentrations of up to 400 µg/mL (Figure 5a) but also similar results regarding antiscratching activity. It also increased both migration (Figure 5b,c) and cell proliferation rates (Figure 5d) of scratched keratinocytes as well as the expression of migration-related factors such as K6, K16, K17 (Figure 5e), and KGF (Figure 5i) and wound-healing-promoting cytokines such as IL-1, IL-17, and IL-22 (Figure 5f–h) in a dose-dependent manner. The EGFR signaling pathway results also showed an increase in the activation of PI3K and AKT proteins (Figure 5j), thus confirming the positive effects of loliolide on wound healing. According to both our loliolide and Pj–EE results on skin aging and wound healing processes as summarized Figure 6, we can conclude that because of its antiaging, antioxidant, anti-inflammatory, antiapoptotic, and wound-healing effects, loliolide holds great potential as a wrinkle-improving and wound-healing compound in the cosmeceutical and medical industries. Further research to determine its effects in the treatment of other skin diseases is needed.

## 4. Materials and Methods

### 4.1. Materials

Loliolide (purity: 98% by HPLC) was purchased from Chemfaces (Wuhan, China). HaCaT cells, a spontaneously transformed aneuploidy immortal keratinocyte cell line from adult human skin, were purchased from the American Type Culture Collection (Rockville, MD, USA). MTT, fetal bovine serum (FBS), phosphate-buffered saline (PBS), penicillin, and Dulbecco’s modified Eagle’s medium (DMEM) were purchased from Gibco (Grand Island, NY, USA). H_2_DCF–DA and the Annexin V-FITC Apoptosis Detection Kit were purchased from Sigma-Aldrich Chemical Co. (St. Louis, MO, USA). TRIzol reagent was purchased from Thermo Fisher Scientific (Waltham, MA, USA). The primer sets for the polymerase chain reaction were synthetized by Macrogen (Seoul, Korea), while PCR premix was purchased from Bio-D Inc. (Seoul, Korea). Primary antibodies to phospho-, total or cleaved forms of PI3K (PI3K; CST #4292, and p-PI3K; CST #4228), AKT (AKT; CST #9272, and p-AKT; CST #4058), ERK (ERK; CST #4696, and p-ERK; CST #9101), caspase 3 (procaspase 3; CST #9665, and cleaved caspase 3; CST #9664), caspase 8 (procaspase 8; CST #4790, and cleaved caspase 8; CST #9496), caspase 9 (procaspase 9; CST #12827, and cleaved caspase 9; CST #7237), and β-actin (CST #4967), and horse radish peroxidase (HRP)-labeled secondary antibodies to antimouse and antirabbit IgG were obtained from Cell Signaling Technology (Danvers, MA, USA).

### 4.2. Extraction of Pj–EE

*Prasiola japonica* used in the experiment was supplied by the *Prasiola japonica* Research Center in Samcheok City, Gangwon-do, Korea. Samples were cut into 2 × 2 cm^2^ pieces and then extracted with 70% ethanol at room temperature for 24 h. Samples and solvents were extracted at a gGa ratio of 1:20 (*w*/*v*). Following extraction, the filtrate was filtered through a 110-nm filter paper (no. 2, Advantec; Toyo Co., Tokyo, Japan), and the filtrate was concentrated using a vacuum concentrator (Eyela New Rotary Vacuum Evaporator, Rikakikai Co., Tokyo, Japan). The concentrated samples were dried using a vacuum freeze dryer (Eyela FD1, Rikakikai Co.) and the yield of the dried samples was measured. The final weight of the extract was 2.752 g (original sample: 44.87 g), with a yield of 6.13%. The dried samples were stored in a −20 °C freezer until use. Analysis of loliolide in Pj–EE was performed by gas chromatography, as reported previously. (https://patents.google.com/patent/KR101591401B1/ko).

### 4.3. Cell Culture

HaCaT cells (a human keratinocyte cell line) were cultured in DMEM supplemented with 10% FBS and 1% penicillin-streptomycin at 37 °C in a 5% humidified incubator.

### 4.4. Cell Viability Assay

HaCaT cells were seeded at a density of 4 × 10^4^ cells per well in a 96-well plate for 24 h and then treated with loliolide (0–100 µM) and Pj–EE (0–400 µg/mL) for 24 h. For testing UV-protective activity, HaCaT cells were pretreated with loliolide (0–100 µM) and then further exposed to UVB radiation (30 mJ/cm^2^). The viability of HaCaT cells treated with loliolide or Pj–EE or UV-irradiated HaCaT cells pretreated with loliolide was measured using the MTT assay, in which cells were first incubated with 10 µL/well of MTT solution (Sigma-Aldrich Chemical Co.) for three hours and then treated with 100 µL of MTT-stopping solution (10% sodium dodecyl sulfate with 10% HCl). After eight hours, the absorbance of the solubilized formazan was measured at 570 nm using an optical density reader (BioTek, Winooski, VT, USA). For determining the compound’s cytotoxicity, the same method was applied using only PJ–EE (0–100 µg/mL) for 24 h [32,33].

### 4.5. ROS Generation Assay (H2DCF-DA Staining)

HaCaT cells were seeded at a density of 4 × 10^5^ cells/mL in a 12-well plate containing previously sterilized, round glass cover slips. After 24 h, cells were treated with loliolide (0–100 µM) for 30 min, washed with PBS, and exposed to UVB (30 mJ). To quantify ROS generation, cells were stained with H_2_DCF–DA (Sigma-Aldrich Chemical Co.) (10 µg/mL) for 30 min in dark conditions. Cells were washed with PBS twice, fixed with 1 mL of 3.7% paraformaldehyde in PBS for 10 min, and the cover slip was then mounted on a rectangular glass slide using mounting solution and left to dry at room temperature for 24 h. Samples were examined using a Nikon Eclipse Ti fluorescence microscope [34] (Nikon, Tokyo, Japan). The florescence intensity of ROS generating cells in three different pictures was quantified using ImageJ and afterwards, the data were summarized in a plot.

### 4.6. DAPI Staining

HaCaT cells were seeded at a density of 4 × 10^5^ cells/mL in a 12-well plate containing previously sterilized, round cover glass slips. After 24 h, cells were treated with loliolide (0–100 µM) for 30 min, washed with PBS, and exposed to UVB (30 mJ). Cells were washed twice with PBS and fixed with 1 mL of 3.7% paraformaldehyde in PBS for 10 min. Cells were washed with PBS two more times, stained with DAPI reagent (Sigma-Aldrich Chemical Co.) (1 µL/mL) for 30 min, and then washed with PBS two more times. The cover slip was then mounted on a rectangular glass slide using mounting solution and left to dry at room temperature for 24 h [35]. Samples were subsequently examined using a Nikon Eclipse Ti fluorescence microscope (Nikon, Tokyo, Japan). The number of damaged cells in three different pictures was quantified using ImageJ, and afterwards the data were summarized in a plot.

### 4.7. Propidium Iodide Staining (FACS)

Apoptosis was analyzed by flow cytometry after different cell treatments. Cells were pretreated with or without loliolide (0–100 µM) and then further irradiated with UVB (30 mJ/cm^2^). For staining, cells were washed twice with cold PBS and resuspended in 1× binding buffer at a concentration of 10^6^ cells/mL. Afterwards, 100 μL of suspension (10^5^ cells) was transferred to E-tubes, 10 μL of propidium iodide and 5 μL of fluorescein isothiocyanate–Annexin V (Sigma-Aldrich Chemical Co.) were added, and cells were incubated for 15 min at room temperature in the dark. Finally, 400 μL of 1× binding buffer (Sigma-Aldrich Chemical Co.) was added and fluorescence was assessed using a Guava easyCyte flow cytometer (Millipore, Burlington, MA, USA).

### 4.8. Western Blot Analysis

HaCaT cells were seeded in a six-well plate (5 × 10^5^ cells/mL) and cultured in a 5% CO_2_ incubator at 37 °C for 24 h. Cells were treated with loliolide (0–100 µM) or Pj–EE (0–200 µg/mL) and incubated for 24 h at 37 °C in a 5% CO_2_ incubator. The cells were then washed three times with cold PBS and lysed with lysis buffer (20 mM of Tris-HCl, pH: 7.4, 2 mM of EDTA, 2 mM of ethylene glycoltetraacetic acid, 50 mM of β-glycerophosphate, 1 mM of orthovanadate, 1 mM of dithiothreitol, 1% Triton X-100, 10% glycerol, 10 µg/mL of aprotinin, 10 µg/mL of pepstatin, 1 mM of benzamidine, and 2 mM of PMSF). The lysates were centrifuged at 12,000 *g* for eight min, and the supernatant was transferred to another tube. Protein quantification was performed by Bradford analysis of the supernatant. Cell lysates (supernatant) quantified by Bradford analysis were analyzed by Western blot. Phosphorylation or the total levels of PI3K (CST #4292, 1:2000), p-PI3K (CST #4228, 1:2000), AKT (CST #9272, 1:2000), p-AKT (CST #4058, 1:2000), and β-actin (CST #4967, 1:2000), and cleaved or total levels of cleaved caspase 3 (CST #9664. 1:2000), procaspase 3 (CST #9665, 1:2000), cleaved caspase 8 (CST #9496, 1:2000), procaspase 8 (CST #4790, 1:2000), cleaved caspase 9 (CST #7237, 1:2000), and procaspase 9 (CST #12827, 1:2000) were evaluated in HaCaT cells. All Western blot data in this study are representative of two experiments showing similar patterns.

### 4.9. Semiquantitative PCR and Quantitative (Real-Time) PCR

HaCaT cells were seeded at 6 × 10^5^ cells per mL in a six-well plate and incubated for 24 h at 37 °C in a 5% CO_2_ incubator. Then, cells were pretreated with loliolide (0–100 µM) or Pj–EE (0–200 µg/mL). The cells were then exposed by UVB (30 mJ/cm^2^) and further incubated for 24 h. TRIzol reagent was used to extract mRNA according to the manufacturer’s instructions [36]. The concentration of the extracted mRNA was measured using a spectrophotometer and the mRNA was used to synthesize complementary DNA (cDNA). cDNA synthesis was performed using a cDNA synthesis kit (Sigma-Aldrich Chemical Co.). Gene-specific primers were designed, as reported previously [37]. RT-PCR and real-time PCR were performed using specific forward and reverse primers (4M 5’- and 3’-primer/reaction) (Table 1 and Table 2).

### 4.10. Cellular Scratching Assay

HaCaT cells were seeded in a six-well plate (5 × 10^5^ cells/mL) and cultured in a 5% at 37 °C in a CO_2_ incubator for 24 h. Scratch wounds were created mechanically with a sterile pipette tip (Ø  =  0.1 mm), whose size range varied from approximately 0.5 mm to 0.9 mm in width. Detached cells and debris were washed away with PBS solution, and fresh medium was treated to the wells. Cells were treated with loliolide (0–100 µM) or Pj–EE (0–200 µg/mL) and incubated for 24 h.

### 4.11. Statistical Analysis

All data are presented as the mean ± standard deviation of at least three independent experiments. A Kruskal–Wallis/Mann–Whitney U test was used to compare statistical differences between the experimental and control groups. A *p*-value of <0.05 was considered to be statistically significant. All statistical analyses were conducted using the Statistical Package for the Social Sciences program (IBM Corp., Armonk, NY, USA).

## Figures and Tables

**Figure 1 ijms-20-00651-f001:**
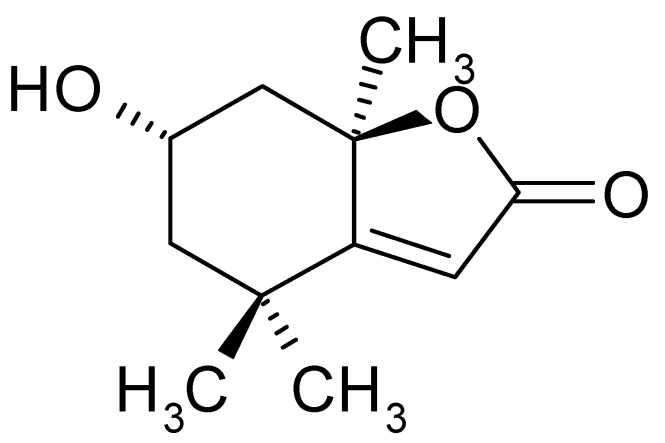
Structure of loliolide.

**Figure 2 ijms-20-00651-f002:**
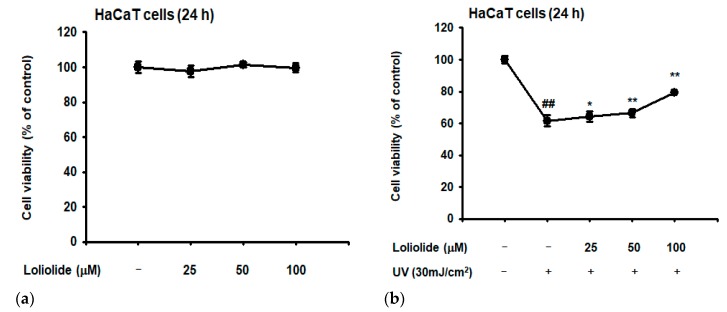
Antiapoptotic and antioxidant effects of loliolide in HaCaT cells. (**a**,**b**) Viability of HaCaT cells after being treated with loliolide alone or loliolide plus UVB, as measured by 3-(4,5-dimethylthiazole-2-yl)-2,5-diphenyltetrazolium bromide (MTT) assay. (**c**) Reactive oxygen species (ROS) generation in H_2_DCFDA-stained HaCaT cells treated with UVB and loliolide was analyzed by confocal microscopy and quantified by calculation of H2DCF-DA intensity signal using ImageJ. (**d**) DAPI staining in HaCaT cells treated with UVB and loliolide. (**e**) FACS analysis in HaCaT cells treated with UVB and loliolide. (**f**) Western blot analysis of the expression of proteins in the apoptosis signaling pathway in UVB- and/or loliolide-treated HaCaT cells. Statistical significance was evaluated using the Kruskal–Wallis/Mann–Whitney U test. ## *p* < 0.05 compared with the normal group (a: Loliolide− or b, c and d: UV−/Loliolide−) and * *p* < 0.05 and ** *p* < 0.01 compared with the control group (b, c, and d: UV+/Loliolide−).

**Figure 3 ijms-20-00651-f003:**
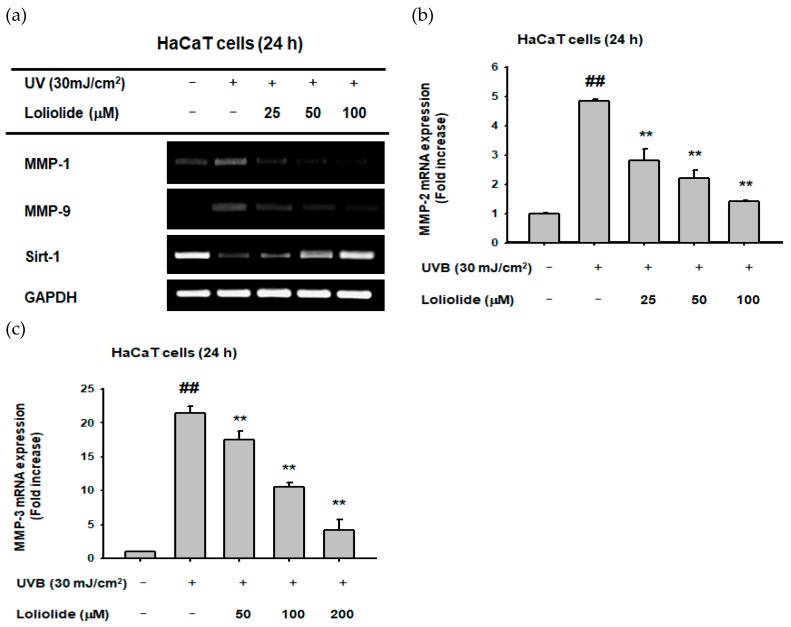
Effect of loliolide on matrix metalloproteinase (MMP) and SIRT-1 expression in HaCaT cells. (**a**) Semiquantitative analysis (RT-PCR) of MMP-1, MMP-9, and SIRT-1 and (**b**,**c**) quantitative analysis of MMP-2 and MMP-3 gene expression in UVB (30 mJ/cm^2^)-irradiated HaCaT cells pretreated with loliolide were performed as described in the Materials and Methods section. Statistical significance was evaluated using the Kruskal–Wallis/Mann–Whitney U test. ## *p* < 0.05 compared with the normal group (b and c: UV−/Loliolide−) and ** *p* < 0.01 compared with the control group (b and c: UV+/Loliolide−).

**Figure 4 ijms-20-00651-f004:**
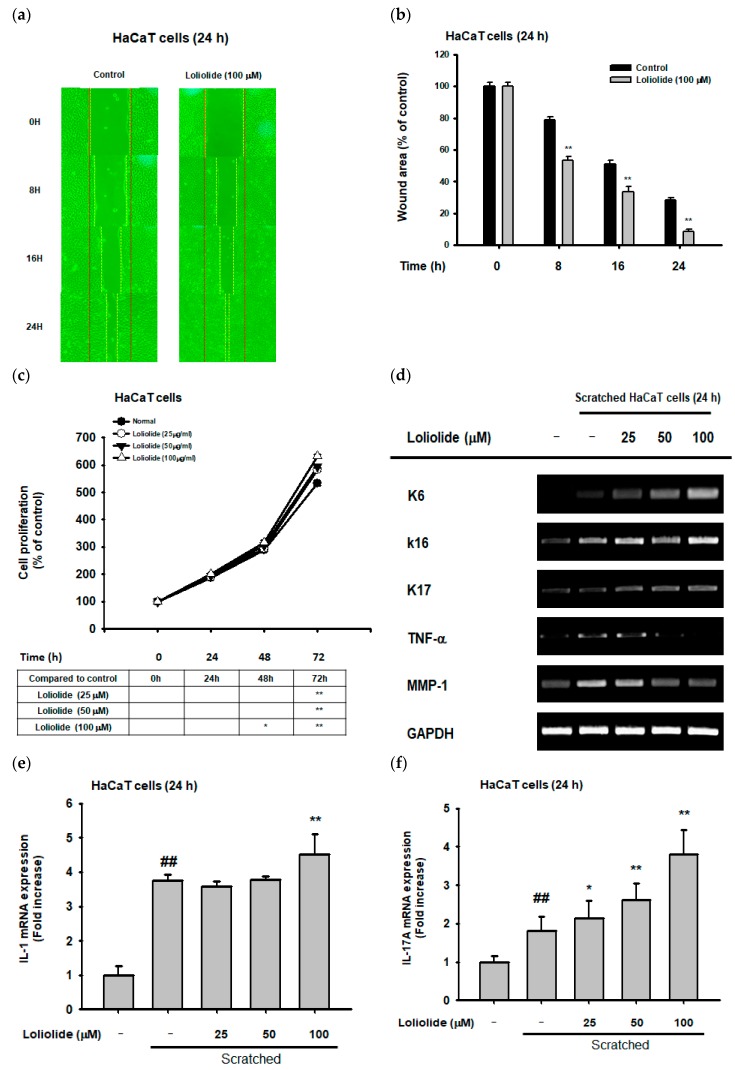
Effect of loliolide on cellular scratching, cell migration and inflammatory genes. (**a**,**b**) Cell migration in loliolide-pretreated scratched HaCaT cells. (**c**) Cell proliferation in loliolide-pretreated HaCaT cells. (**d**–**h**) RT-PCR (**d**) and real-time PCR (**e**–**h**) analyses of migration and inflammatory gene expression in loliolide-pretreated HaCaT cells. (**i**) Western blot analysis of the expression of epidermal growth factor receptor (EGFR) signaling pathway proteins in loliolide-pretreated scratched HaCaT cells. Statistical significance was evaluated using the Kruskal–Wallis/Mann–Whitney U test. ## *p* < 0.05 compared with the normal group (e–h: Loliolide−/Scratched−), and * *p* < 0.05 and ** *p* < 0.01 compared with the control group (b and c: Loliolide−, and f, g, and h: Loliolide−/Scratched+).

**Figure 5 ijms-20-00651-f005:**
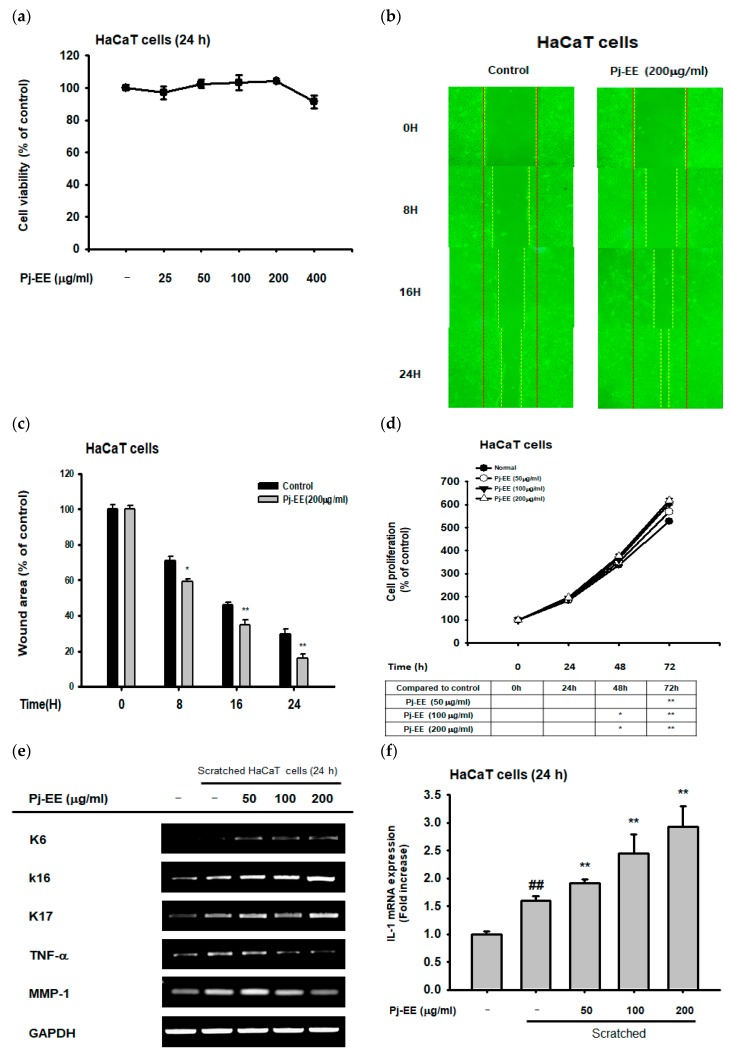
Effects of Pj–EE on cellular scratching, cell migration, and inflammatory genes. (**a**) Measurement of cell viability in Pj–EE-pretreated and scratched HaCaT cells. (**b**,**c**) Cell migration in Pj–EE-pretreated scratched HaCaT cells. (**d**) Cell proliferation in HaCaT cells treated with Pj–EE. (**e**–**i**) RT-PCR and real-time PCR analyses of migration and inflammatory gene expression in Pj–EE-treated HaCaT cells. (**j**) Western blot analysis of the expression of the EGFR signaling pathway in Pj–EE-pretreated scratched HaCaT cells. Statistical significance was evaluated using the Kruskal–Wallis/Mann–Whitney U test. ## *p* < 0.05 compared with the normal group (f–i: Loliolide−/Scratched−), and * *p* < 0.05 and ** *p* < 0.01 compared with the control group (c and d: Pj–EE−, and f–i: Loliolide−/Scratched+).

**Figure 6 ijms-20-00651-f006:**
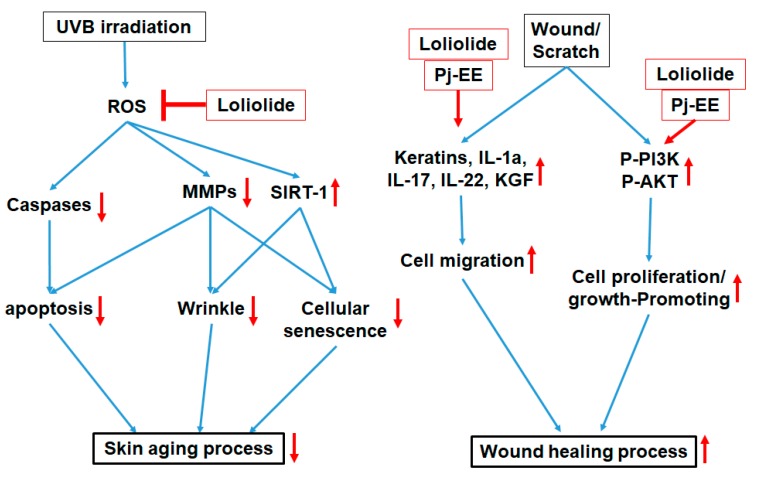
Antiapoptotic, antioxidant, and wound-healing effects of loliolide and Pj–EE in HaCaT keratinocytes.

**Table 1 ijms-20-00651-t001:** Sequences of primers (human) used in semiquantitative RT-PCR.

Name	Primer	Sequence (5’ to 3’)
MMP-1	Forward	TCTGACGTTGATCCCAGAGAGCAG
	Reverse	CAGGGTGACACCAGTGACTGCAC
MMP-9	Forward	GCCACTTGTCGGCGATAAGG
	Reverse	CACTGTCCACCCCTCAGAGC
K6	Forward	GAGCGGCCATGAAGAAGCT
	Reverse	TCCGCCATGCACCAACTTA
K16	Forward	CTGAGCCGCATCCTGAATGA
	Reverse	TCGCGGGAAGAATAGGATTGG
K17	Forward	CATGCAGGCCTTGGAGATAGA
	Reverse	CACGCAGTAGCGGTTCTCTGT
SIRT-1	Forward	CAGTGTCATGGTTCCTTTGC
	Reverse	CACCGAGGAACTACCTGAT
TNF-α	Forward	GTGACAAGCCTGTAGCCCAT
	Reverse	CAGACTCGGCAAAGTCGAGA
GAPDH	Forward	CACTCACGGCAAATTCAACGGCAC
	Reverse	GACTCCACGACATACTCAGCAC

**Table 2 ijms-20-00651-t002:** Sequences of primers (human) used in quantitative real-time PCR.

Name	Primer	Sequence (5’ to 3’)
MMP-2	Forward	AAAACGGACAAAGAGTTGGCA
	Reverse	CTGGGGCAGTCCAAAGAAC
MMP-3	Forward	TGTTAGGAGAAAGGACAGTGGTC
	Reverse	CGTCACCTCCAATCCAAGGA
IL-1	Forward	CTTCTGGGAAACTCACGGCA
	Reverse	AGCACACCCAGTAGTCTTGC
IL-17	Forward	CGGACTGTGATGGTCAAC
	Reverse	CAAGGTGAGGTGGATCGGTT
IL-22	Forward	AGCCCTATATCACCAACCGC
	Reverse	TCTCCCCAATGAGACGAACG
GAPDH	Forward	GACAGTCAGCCGCATCTTCT
	Reverse	GCGCCCAATACGACCAAATC

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
