# Peer review of "Loliolide Presents Antiapoptosis and Antiscratching Effects in Human Keratinocytes"

_ijms, 2019, doi:10.3390/ijms20030651_

Round 1
Reviewer 1 Report
In the manuscript entitled “Loliolide presents anti-apoptosis and wound-healing effects in human keratinocytes” the authors demonstrate that Loliolide improves markers of wound healing and has anti-oxidant properties. It is a experimentally straight forward and most of the studies include statistical analysis. The authors have some missing information and errors, listed below.
-Please define HaCat cells.
-In Fig 2C, the images are not visible and there are no p values.
-Fig 3 title. Anti-wrinkling effects is not what the figure is demonstrating. Please change title.
-Fig 4b, please add stv dev and p values.
-Fig 4d, in the text the authors talk about MMP-9 but they show MMP-1 on the blot. Same in the discussion. Cox-2 in the Western does not follow what it said in the text. It is down a 25 and then up at 100 and also may be saturated.
-Fig 4e,f,g, the authors say in the text that there is a decrease in IL 1,17,22 but there is an increase but in the discussion the increase is noted.
Fig 5, same problems as above, fig 5b, please add statistics, again, MMP-1 and MMP-9 are not discussed properly, Fig 5e, again, IL is increased but it says decreased.
Author Response
Reviewer #1.
-Please define HaCat cells.
: We have included additional information on this like “a human keratinocyte cell line” in L23, L244.
-In Fig 2C, the images are not visible and there are no p values.
: We have improved visibility of Fig. 2C and also included statistical significance in bottom panel of Fig. 2c like other figure style (see Fig. 2c in p4).
-Fig 3 title. Anti-wrinkling effects is not what the figure is demonstrating. Please change title.
: We have changed the title to “Effect of loliolide on MMP expression” in L118 and 127 of p6.
-Fig 4b, please add stv dev and p values.
: We have changed the figure with mean and STD value and also included statistical significance like other figures (see Fig. 4b in p7).
-Fig 4d, in the text the authors talk about MMP-9 but they show MMP-1 on the blot.
: We have changed it to MMP-1 in text (see L 141, p6).
-Same in the discussion. Cox-2 in the Western does not follow what it said in the text.
: We have deleted it in text except Introduction section.
-It is down a 25 and then up at 100 and also may be saturated.
: We also think the effects like this in cases of some parameters.
-Fig 4e,f,g, the authors say in the text that there is a decrease in IL 1,17,22 but there is an increase but in the discussion the increase is noted.
: We have fixed this like “while IL-1, IL-17, and IL-22 genes were dose-dependently increased (Figures 4e, 4f, and 4g)” in L144 of p6.
-Fig 5, same problems as above, fig 5b, please add statistics,
: We have included statistical significance for wound healing data (Fig. 5C).
again, MMP-1 and MMP-9 are not discussed properly, Fig 5e, again, IL is increased but it says decreased.
: We have fixed this like “decrease in the expression of the inflammation-related genes MMP-1 and TNF-a (Figure 5e); while there is an important increase in IL-1, IL-17, and IL-22 in a dose-dependent manner (Figures 5f, 5g, and 5h)” in L171-173 of p9.
Reviewer 2 Report
The authors have looked at the role of loliolide on human keratinocyte activities. Those studies are done well but the conclusions are overstated and need to clarified. For instance, the authors conclude that their studies suggest improvements in wound healing but the only studies performed are the "scratch" keratinocyte migration study that is just a small component of wound healing. You then jump to the conclusion that you have improved wrinkling but there is no evidence to support changes in wrinkle formation. There are a few other issues.
1) I do not believe that anything improves normal wound healing.
2) The statement that wound healing is triggered by EGF signaling is not completely true.
3) Expression of cytokines can be both good and bad on wound healing depending on the extent of expression.
4) All of the studies use HaCaT cells that are a keratinocyte cell line. Are they transformed (cancer cells) that have different reactions than normal epithelial cells?
5) Several figures, especially 2c are almost impossible to see any difference. The epithelial scratch studies are also very difficult to see.
6) You claim significant differences for some studies (4b, 4c, 5c) but to my eye they do not look to be statistically significant.
Author Response
Reviewer #2.
The authors have looked at the role of loliolide on human keratinocyte activities. Those studies are done well but the conclusions are overstated and need to clarified. For instance, the authors conclude that their studies suggest improvements in wound healing but the only studies performed are the "scratch" keratinocyte migration study that is just a small component of wound healing. You then jump to the conclusion that you have improved wrinkling but there is no evidence to support changes in wrinkle formation.
: This is good comment. So, we have changed anti-wrinkle effect to antiscratching effect and cellular scratching to avoid overstatement in our paper’s conclusions. You can find all changes marked with yellow painting (see L32, 33, 84, 137, 151, 161, 165, 176, 224, 231, 232, and 341).
There are a few other issues.
1) I do not believe that anything improves normal wound healing.
: Based on Fig. 4a and 4c, loliolide displayed enhancement of cell scratched conditions and increase of cell proliferation. Although we need animal experiment to confirm this, our data strongly suggest that this compound has putative antiscratching effect. Since scratching is one of well-acceptable wound healing tests, we would like to assume that it has wound healing activity. However, we also agree with above comment, we would like to express only like antiscratching or cellular scractching model rather than wound healing effect anti-wound healing.
2) The statement that wound healing is triggered by EGF signaling is not completely true.
: It is correct, but EGF/EGFR-induced signaling pathway is also considered as one of major wound-healing pathways. Thus, a mention that EGFR activation occurred shortly after scratch wounding in renal tubular cells. Wound repair after scratching was significantly promoted by EGF and suppressed by EGFR inhibitor gefitinib (Am J Physiol Renal Physiol. 2017 Jun 1;312(6):F963-F970) still led us to consider that EGF/EGFR signaling is important for would repairing. So, we have little corrected it like “One of major wound-healing processes is triggered by the activation of the epidermal growth factor receptor (EGFR) pathway”.
3) Expression of cytokines can be both good and bad on wound healing depending on the extent of expression.
: This is good point. We also agree with this and added this point in L65-70 like “Wound healing begins with the induction of an inflammatory response due to bacterial colonization and barrier disruption, which stimulate the immediate production of interleukin (IL)-1, tumor necrosis factor alpha (TNF-a), and Th17-produced cytokines [17]. Th17 cell–associated cytokines such as IL-17 and IL-22 [18] can impact wound healing positively by clearing pathogens and modulating mucosal surfaces and epithelial cells [19], although these cytokines sometimes negatively function in wound healing process by inducing cellular damages”.
4) All of the studies use HaCaT cells that are a keratinocyte cell line. Are they transformed (cancer cells) that have different reactions than normal epithelial cells?
: HaCaT cells are a spontaneously transformed aneuploid immortal keratinocyte cell line from adult human skin, widely used in scientific research. HaCaT cells are utilized for their high capacity to differentiate and proliferate in vitro. Their use in research allows for the characterization of human keratinocyte using a model that is reproducible and addresses issues such as short culture lifespan and variations between cell lines that would otherwise be encountered. Although the cells are immortal cells, it is known that the cells show similar cellular and molecular patterns to normal skin keratinocytes. So far, thus, 4,871 papers prepared working with HaKaT cells have published. Relevant sentence has been included in L245 like “a spontaneously transformed aneuploidy immortal keratinocyte cell line from adult human skin”.
5) Several figures, especially 2c are almost impossible to see any difference. The epithelial scratch studies are also very difficult to see.
: We have improved the quality of Fig. 2c and Fig. 4a, 4b.
6) You claim significant differences for some studies (4b, 4c, 5c) but to my eye they do not look to be statistically significant
: We have clearly observed statistical significance, according to our calculation.
Reviewer 3 Report
The manuscript describes the antiapoptosis and wound-healing effects of loliolide and the loliolide-rich Prasiola japonica ethanol extract in human keratinocytes. The protective effect of loliolide in ultraviolet-damaged keratinocytes is also demonstrated.
This is an interesting study in which the authors provide numerous, clear and significant results to show the antiapoptotic, antioxidant and antiwrinkling, together with wound-healing and cell migration-promoting effects of loliolide, while no cytotoxicity at the employed doses is reported. The study is of interest to the readers and provides data that support the potential of loliolide as a wound-healing compound that could be further investigated for its use in the cosmeceutical industry.
The manuscript is well-written with appropriate language and a clear research question. The study is correctly designed and shows conclusive results. The discussion does not over-interpret the results obtained. Relevant and current literature is well considered.
However, there are some points that need to be addressed, corrected and/or clarified.
1) Introduction. Page 2, lines 77-78. Please include a reference supporting the loliolide-rich content of the Prasiola japonica ethanol extract.
2) Figures 2, 4 and 5 are too long to be easily followed. The understanding could be improved by splitting them in further figures so that the figure caption could appear closer to the corresponding figures.
3) The different study groups (normal, control) should be clearly described in the Materials and Methods section and accordingly considered in figures and the text. In this regard, if the control group consists of UVB-exposed cells in figure 2, could the Y-axis in figures 2a and 2b state Cell viability (% of normal cells) to not to be confused with the control group (loliolide -/UV +) showed in figure 2b?
4) Page 3, line 90.
It is stated that experiments are performed in loliolide-pretreated cells, which were subjected to UVB radiation. However, in the Materials and Methods section it is described that cells were exposed to UVB radiation and then treated with loliolide (page 13, line 251) in the cell viability assay. Please, clarify the order in loliolide and UVB treatment.
5) Page 4, Figures 2c and 2d. Please include magnification or scale bar in the immunofluorescence images.
6) Please mention the Kruskal-Wallis statistical test in the Materials and Methods section (Statistical analysis section, page 14, line 315) according to the figures captions (Figure 2, page 5, line 109; Figure 4, page 8, line 144 and Figure 5, page 10, line 167).
7) Figure 3 caption. Page 6, line 121. Please rephrase to clearly specify that: (a) Semiquantitative analysis (RT-PCR) of MMP-1, MMP-9 and SIRT-1 and (b,c) quantitative analysis of MMP-2 and MMP-3 gene expression in HaCaT cells …
8) Figure 3 caption. Page 6, line 122. Please include the statistical significance meaning related to ## and **.
9) Some references to figures are altered in the text, please correct them:
- Page 6, line 132. KGF (g) should be changed to KGF (Figure 4h) according to the figure shown.
- Page 6, line 134. In a similar way, 4h should be changed to 4g.
- Page 8, line 157. (Figure 5h) should be (Figure 5i).
- Page 8, line 159. 5i should be 5h.
- Page 12, line 216. (Figure 5h) should be changed to (Figure 5i).
- Page 12, line 217. 5i should be changed to 5h.
10) Page 6, line 133. The inflammation-related gene MMP-9 reported in the text is not included in figure 4d, but MMP-1 is. Please, clarify if the correct information is in the text, in the figure or if some information about one of the metalloproteases is missing (since both sequences appear in table 1) and change accordingly.
11) Page 8, line 158. As in the previous item, the inflammation-related gene MMP-9 reported in the text is not included in figure 5e, but MMP-1 is. Please, correct accordingly.
12) Page 6, line 132-134. Please revise the sentence. In its current form it states that loliolide showed a remarkable decrease in the expression of IL-1, IL-17 and IL-22 while the opposite is shown in figures 4e, 4f and 4g.
13) Page 8, lines 157-159. As in the previous item, please revise the sentence. It states that loliolide showed a remarkable decrease in the expression of IL-1, IL-17 and IL-22 while the opposite is shown in figures 5f, 5g and 5h.
14) Figure 5 caption. Page 10, line 168. Please include the statistical significance for *, since it appears in figures 5g and 5h.
15) Page 11, lines 185-187. Please revise the sentence for a possible typo error, since the meaning is not completely clear.
16) Extraction of Pj-EE. Page 12, lines 236-245. Please comment and include the following information:
- Analysis performed to demonstrate the presence of loliolide in the specific Prasiola japonica ethanol extract used (HPLC, TLC,…) or if the supplier provided these data.
- If quantification of loliolide has been performed in this extract to assure that it is a loliolide-rich source.
- Are data about the HPLC profiling of the extract available to exclude that other possible components present could be affecting its effect?
17) Page 13, line 251. The range of concentrations of Pj-EE employed goes from 0-200 µg/mL, however the figure 5a regarding cell viability goes up to 400 µg/mL. Please, correct accordingly.
18) Page 13, line 261. Please report additional data (manufacturer,…) regarding H2DCF-DA employed in the ROS generation assay.
19) Page 13, line 265. Please further explain how quantification to get the graph in figure 2c was performed (by microscopy, by a microplate reader, if the % relates to the number of positive cells or intensity of staining…).
20) Page 13, lines 280-282. Please include further information regarding the manufacturer of the different reactives used (fluorescein isothiocyanate-Annexin V, 1x binding buffer…).
21) Page 13, line 291. In order to increase the reproducibility of the experiments by other researchers, please express the acceleration to which samples are subjected during centrifugation in g instead of the angular speed applied in rpm, since the acceleration is related to the radius of the specific centrifuge employed and rpm don´t allow to get the same acceleration if a centrifuge with a different radius is used.
22) Page 14, lines 294-295. Please include further information regarding the antibodies employed in Western blot analysis (manufacturer/supplier, clone, reference number,…) as well as the dilution used for each of them.
23) Page 14, line 304. Please include additional information about the cDNA synthesis kit used (reference, manufacturer, enzyme…) used in PCR analyses.
24) Page 14, line 305. Please confirm that the design of the gene-specific primers was previously reported in reference 33.
25) Page 14, line 306. The sequences of Sirt-1, TNF-α and COX-2 are missing in table 1 (page 15), please include them.
26) In reference 22, authors are missing. Please include the missing information.
Author Response
Reviewer #3.
However, there are some points that need to be addressed, corrected and/or clarified.
1) Introduction. Page 2, lines 77-78. Please include a reference supporting the loliolide-rich content of the Prasiola japonica ethanol extract.
We have included previous references in this manuscript like below
Park S.H.; Choi E.; Kim S.; Kim D.S.; Kim J.H.; Chang S.; Choi J.S.; Park K.J.; Roh K.B.; Lee J.; Yoo B.C.; Cho J.Y. Oxidative Stress-Protective and Anti-Melanogenic Effects of Loliolide and Ethanol Extract from Fresh Water Green Algae, Prasiola japonica. Int J Mol Sci 2018, 19.
Sungsoo Joo, Patent No: 10-2014-0026927, Pharmaceutical composition for treating cancer. 2016.01.28. [https://patents.google.com/patent/KR101591401B1/ko]
2) Figures 2, 4 and 5 are too long to be easily followed. The understanding could be improved by splitting them in further figures so that the figure caption could appear closer to the corresponding figures.
: Although this is good point, we would like to keep current format, since all data are related in each figure.
3) The different study groups (normal, control) should be clearly described in the Materials and Methods section and accordingly considered in figures and the text. In this regard, if the control group consists of UVB-exposed cells in figure 2, could the Y-axis in figures 2a and 2b state Cell viability (% of normal cells) to not to be confused with the control group (loliolide -/UV +) showed in figure 2b?
: We have indicated these in L114-117 like “## p < 0.05 compared with the normal group (a: Loliolide - or b, c, and d: UV - / Loliolide -) and ** p < 0.01 compared with the control group (b, c, and d: UV + / Loliolide -)”, in L130-132 like “## p < 0.05 compared with the normal group (b and c: UV - / Loliolide -) and ** p < 0.01 compared with the control group (b and c: UV + / Loliolide -)”, in L155-158 like “## p < 0.05 compared with the normal group (e, f, g, and h: Loliolide - / Scratched -), and * p < 0.05 and ** p < 0.01 compared with the control group (e, f, g, and h: Loliolide - / Scratched +)”, and in L181-184 like “## p < 0.05 compared with the normal group (f, g, h, and i: Loliolide - / Scratched -), and * p < 0.05 and ** p < 0.01 compared with the control group (c and d: Pj-EE -, and f, g, h, and i: Loliolide - / Scratched +).
4) Page 3, line 90. It is stated that experiments are performed in loliolide-pretreated cells, which were subjected to UVB radiation. However, in the Materials and Methods section it is described that cells were exposed to UVB radiation and then treated with loliolide (page 13, line 251) in the cell viability assay. Please, clarify the order in loliolide and UVB treatment.
: For this experiment, we pretreated loliolide 30 min before and checked viability after irradiation. We have fixed the sentences to make it clear like “In contrast, the viability of HaCaT cells was significantly decreased under UVB irradiation (30 mJ/cm2) condition (Figure 2b). However, pretreated loliolide (100 mM) clearly recovered the down-regulated level of cell viability of UV-irradiated HaCaT cells, implying a putative photoprotective effect against cell death caused by UVB-induced oxidative stress (Figure 2b)” in L93-96 of p3.
5) Page 4, Figures 2c and 2d. Please include magnification or scale bar in the immunofluorescence images.
: We have improved the quality of these figures and therefore now you can easily discriminate the difference between UV alone and loliolide-pretreated UV-irradiation cell groups.
6) Please mention the Kruskal-Wallis statistical test in the Materials and Methods section (Statistical analysis section, page 14, line 315) according to the figures captions (Figure 2, page 5, line 109; Figure 4, page 8, line 144 and Figure 5, page 10, line 167).
: We have mentioned it in L347 like “A Kruskal–Wallis/Mann–Whitney U test” as well as Figure legends in Fig. 3, 4, and 5 (L115, L131, L156, and L181).
7) Figure 3 caption. Page 6, line 121. Please rephrase to clearly specify that: (a) Semiquantitative analysis (RT-PCR) of MMP-1, MMP-9 and SIRT-1 and (b,c) quantitative analysis of MMP-2 and MMP-3 gene expression in HaCaT cells …
: We have rephrased according to this comment like “a) Semiquantitative analysis (RT-PCR) of MMP-1, MMP-9 and SIRT-1, and (b,c) quantitative analysis of MMP-2 and MMP-3 gene expression in UVB (30 mJ/cm2)-irradiated HaCaT cells pretreated with loliolide were performed as described in Materials and methods section” in L127-130 of p6.
8) Figure 3 caption. Page 6, line 122. Please include the statistical significance meaning related to ## and **.
: We have included all in each figure legend.
9) Some references to figures are altered in the text, please correct them:
- Page 6, line 132. KGF (g) should be changed to KGF (Figure 4h) according to the figure shown.
- Page 6, line 134. In a similar way, 4h should be changed to 4g.
- Page 8, line 157. (Figure 5h) should be (Figure 5i).
- Page 8, line 159. 5i should be 5h.
- Page 12, line 216. (Figure 5h) should be changed to (Figure 5i).
- Page 12, line 217. 5i should be changed to 5h.
: We have changed all according to these comments.
10) Page 6, line 133. The inflammation-related gene MMP-9 reported in the text is not included in figure 4d, but MMP-1 is. Please, clarify if the correct information is in the text, in the figure or if some information about one of the metalloproteases is missing (since both sequences appear in table 1) and change accordingly.
: We have included it, which was missed in the context (see L141).
11) Page 8, line 158. As in the previous item, the inflammation-related gene MMP-9 reported in the text is not included in figure 5e, but MMP-1 is. Please, correct accordingly.
: We have included it, which was missed in the context (see L171).
12) Page 6, line 132-134. Please revise the sentence. In its current form it states that loliolide showed a remarkable decrease in the expression of IL-1, IL-17 and IL-22 while the opposite is shown in figures 4e, 4f and 4g.
: We have fixed to make it clear like “it showed a significant downregulation in the expression of inflammation-related genes MMP-1 and TNF-a (Figure 4d), while IL-1, IL-17, and IL-22 genes were dose-dependently increased (Figures 4e, 4f, and 4g)” in L144-145 of p6.
13) Page 8, lines 157-159. As in the previous item, please revise the sentence. It states that loliolide showed a remarkable decrease in the expression of IL-1, IL-17 and IL-22 while the opposite is shown in figures 5f, 5g and 5h.
: We have fixed to make it clear like “decrease in the expression of the inflammation-related genes, MMP-1 and TNF-a (Figure 5e), while there was an important increase in IL-1, IL-17, and IL-22 in a dose-dependent manner (Figures 5f, 5g, and 5h).” in L171-173 of p9.
14) Figure 5 caption. Page 10, line 168. Please include the statistical significance for *, since it appears in figures 5g and 5h.
: We have included it (see L183).
15) Page 11, lines 185-187. Please revise the sentence for a possible typo error, since the meaning is not completely clear.
: We have revised this part like “In addition to presenting a good antiapoptotic effect, loliolide also dose-dependently reduced not only oxidative stress (Figure 2d), but also DNA damage (Figure 2e), cell death (Figure 2f), and apoptosis-related caspase activation (Figure 2g) exhibited under UV irradiation conditions” in L202-204 of p11.
16) Extraction of Pj-EE. Page 12, lines 236-245. Please comment and include the following information:
- Analysis performed to demonstrate the presence of loliolide in the specific Prasiola japonica ethanol extract used (HPLC, TLC,…) or if the supplier provided these data.
- If quantification of loliolide has been performed in this extract to assure that it is a loliolide-rich source.
- Are data about the HPLC profiling of the extract available to exclude that other possible components present could be affecting its effect?
: We have included related sentence in L268-270 like “Analysis of loliolide in Pj-EE was performed by gas chromatography, as reported previously [https://patents.google.com/patent/KR101591401B1/ko]”.
17) Page 13, line 251. The range of concentrations of Pj-EE employed goes from 0-200 µg/mL, however the figure 5a regarding cell viability goes up to 400 µg/mL. Please, correct accordingly.
: Since it is cell viability test, we have tested it up to 400 µg/ml concentration, although all our experiments have done with 200 mg/ml. Therefore, it was found that Pj-EE is very non-toxic until 400 µg/ml.
18) Page 13, line 261. Please report additional data (manufacturer,…) regarding H2DCF-DA employed in the ROS generation assay.
: We have included the manufacturer in L248 and L289.
19) Page 13, line 265. Please further explain how quantification to get the graph in figure 2c was performed (by microscopy, by a microplate reader, if the % relates to the number of positive cells or intensity of staining…).
: We have included the method to quantify like “The number of damaged cells in three different pictures was quantified using ImageJ and afterwards the data was summarized in a plot” in L295-305.
20) Page 13, lines 280-282. Please include further information regarding the manufacturer of the different reactives used (fluorescein isothiocyanate-Annexin V, 1x binding buffer…).
: We have included the manufacturer in L311 and L313.
21) Page 13, line 291. In order to increase the reproducibility of the experiments by other researchers, please express the acceleration to which samples are subjected during centrifugation in g instead of the angular speed applied in rpm, since the acceleration is related to the radius of the specific centrifuge employed and rpm don´t allow to get the same acceleration if a centrifuge with a different radius is used.
: We have included g value in L322.
22) Page 14, lines 294-295. Please include further information regarding the antibodies employed in Western blot analysis (manufacturer/supplier, clone, reference number,…) as well as the dilution used for each of them.
: We have included all of information in L325-329 of p14 like “of PI3K (CST # 4292, 1: 2000), p-PI3K (CST #4228, 1: 2000), AKT (CST #9272, 1: 2000), p-AKT (CST # 4058, 1: 2000), and β-actin (CST # 4967, 1: 2000), and cleaved or total levels of cleaved caspase 3 (CST # 9664. 1: 2000), procaspase 3 (CST # 9665, 1: 2000), cleaved caspase 8 (CST # 9496, 1: 2000), procaspase 8 (CST # 4790, 1: 2000), cleaved caspase 9 (CST # 7237, 1: 2000), and procaspase 9 (CST # 12827, 1: 2000)”.
23) Page 14, line 304. Please include additional information about the cDNA synthesis kit used (reference, manufacturer, enzyme…) used in PCR analyses.
: We have included all of information in L338.
24) Page 14, line 305. Please confirm that the design of the gene-specific primers was previously reported in reference 33.
: We have included correct reference here with [37] related to MMP analysis.
Hong Y.H.; Kim D.; Nam G.; Yoo S.; Han S.Y.; Jeong S.G.; Kim E.; Jeong D.; Yoon K.; Kim S.; Park J.; Cho J.Y. Photoaging protective effects of BIOGF1K, a compound-K-rich fraction prepared from Panax ginseng. J Ginseng Res 2018, 42, 81-89
25) Page 14, line 306. The sequences of Sirt-1, TNF-α and COX-2 are missing in table 1 (page 15), please include them.
: We have included these sequences (see Table 1 of p13).
26) In reference 22, authors are missing. Please include the missing information.
: We have included author’s information like “Yang X.; Kang M.C.; Lee K.W.; Kang S.M.; Lee W.W.; Jeon Y.J”.
Reviewer 4 Report
The submitted manuscript presents the results on an in vitro study into the utility of loliolide (and a natural extract high in this active compound) as a photo protective and wound healing agent. While the data does suggest that this compound may offer protection from UVB damage and stimulates important wound healing processes in keratinocytes and would be of interest to the readers of IJMS and the dermatology and wound healing research communities at large, there are a number errors in the manuscript which must be addressed prior to further consideration for publication.
ROS is an abbreviation which stands for ‘reactive oxygen species’ however, on lines 41-42 the authors describe it as ‘radical oxygen species’. Indeed both the references used be the authors also correctly use the term reactive oxygen species. Please correct this error.
The statement “Prasiola japonica ethanol extract (Pj-EE), is a loliolide-rich source that has also been shown to possess antioxidant activity.” On lines 77-78 requires a reference.
TH17 is written differently in the two instances on line 66.
Many of the figures have too many panels included, which becomes a problem where individual panels are then made too small to be clearly visualised. In particular, the H2DCF-DA staining in figure 2 is much smaller than the DAPI staining (figure 2d) and as such is very difficult to see and make out at all. Consider separating into multiple figures if appropriate, but at least ensure that the information can be clearly seen.
Moreover, there are issues with the labelling of the panels in figure 2. The legend says that (c) is the H2-DCFDA–stained HaCaT cells treated with UVB and loliolide. But this is in fact the quantification graph of ROS generation (assuming that this is made from the staining above which does not have any panel identification. Please include identification of all panels and include description of ‘ROS generation’ quantification in the legend (ie is it intensity of fluorescent signal of H2DCF-DA>) and ensure that the methods are updated also.
Figure legend 2 also states that statistical analysis was performed both comparing to the normal (non-UV treated) and control (UV treated with no loliolide treatment). In that case please include ## in all panels where there are significant differences between the groups compared with the normal group. For example, in 2b where UV treatment clearly reduces cell viability, and likely is still significantly down in the loliolide 25 and 50 groups at least, ## should be included in line with the figure legends description. Stats (or at least significance markers) appear to be entirely missing from the graph in 2c.
The results 2.2 on line 115 mention analysis of gene expression in H2O2-treated HaCaT cells. While this is also mention in the methods on line 300, no justification for using H202 rather than UVB irradiation is given, and significantly, the figure legends and all graphs state that the experiments were carried out in UV treated HaCaTs with or without loliolide treatment. This must be corrected. This sentence should also be reworded to “expression of MMP-1, MMP-9 (Figure 3a), MMP-2 (Figure 3b) and MMP-3 (Figure 3c) as well as restoring Sirt-1 expression (Fig. 3a)”. Please also include stats details in the legend for figure 3.
In the results 2.3 on line 125-126 is states that loliolide showed an increase in cell migration and wound repair in HaCaT cells in a dose-dependent manner (Figure 4a and 4b). However, the figures only present data for a single dose of loliolide (100uM) compared to control. This statement is therefore incorrect and should be changed. A similar statement is made in results 2.4 on line 153 referring to Pj-EE which was only assessed at 200ug/ml for the scratch wound assay. Both Figures (4b and 5c) need error bars and stats included and the images in panels 4a and 5b are very unclear and need to be improved so that the cells can be better visualised.
The results 2.3 and 2.4 also talk about MMP-9 (lines 133 and 158), but the figures (4d and 5e) are labelled MMP-1. Both MMP-1 and MMP-9 primers are listed in table 1 of the methods and were analysed in figure 3a so I do not know what gene was analysed in this section. Also, the primers for KGF are missing. Please correct.
Both of these results section also incorrectly state that loliolide and Pj-EE showed a remarkable decrease in IL-1, IL-17 and IL-22 (lines 133-134 and 158-159) where the figures show that the expression of these markers are clearly increased. The discussion (lines 209-211) correctly states that the expression of these wound healing-promoting cytokines is enhanced. Please correct the results.
I would also suggest changing ‘cell migration and wound repair’ on line 125 to the more correct terms ‘cell migration and scratch wound closure’ and also ensuring that the order of the genes presented in the results (eg line 133 MMP-9, TNFa, COX2) matches the order of the genes in the figures (figure 4d TNFa, MMP-1, COX-2) throughout.
The legend of Figure 4 needs to include ‘and inflammatory genes’ after migration on line 142 to match the data presented and also the legend of figure 5.
Figure 6 is not very useful. Without inclusion of arrows to describe the effect of loliolide and Pj-EE on the genes listed (eg up regulation or down regulation) the flow chart is meaning-less. Perhaps remove this figure entirely and spread out the panels from other figures for easier reading?
Lines 185-187 makes a strange sentence and should be reworded. Also, including mention of figures in the discussion seems strange considering this is already presented in the results.
Please update the methods (line 233-235) to indicate that these are antibodies, list the catalogue number and also include details of the secondary antibodies and detected methods used.
The methods 4.3 states that or determining the compound’s cytotoxicity, the same method was applied using only PJ-EE (0–100 μg/mL) for 24 hours. This appears incorrect as the loliolide was investigated in normal HaCaTs at 0-100uM in figure 2a and Pj-EE at 0-400ug/ml in figure 5a. Please correct.
Methods 4.6 also needs to be expanded to include the treatment time prior to FACS analysis and more details about the order of irradiation and treatment. All the previous experiments were performed with pretreatment followed by irradiation, the methods state loliolide treatment after UVB irradiation (line 277) and the results say is was simultaneous (line 101). Please clarify.
Please included the quantity of template used in the RT-PCR analysis (methods 4.8) and also clarify why two different methods for analysing gene expression (gels versus fold change) were used.
Methods 4.9 ends by stating that debris was removed by PBS. Please expand and included details about whether normal culture media or treated media was replaced in the wells.
Finally, what is the concentration of loliolide compound within the Pj-EE extracts at the concentrations presented? Is it likely that the 200ug/ml Pj-EE dose is delivering an equivalent amount of loliolide at the effective dose of 100uM or could it be higher? This information should be included. Also, why were the UV-protective effects of Pj-EE not investigated? The methods 4.3 state that UVB radiation was performed and Pj-EE treatment at 0-200 ug/ml performed but I can’t see this in the results. Only scratch data is presented in figure 5. Can this also be performed and included in the manuscript?
Author Response
Reviewer #4. While the data does suggest that this compound may offer protection from UVB damage and stimulates important wound healing processes in keratinocytes and would be of interest to the readers of IJMS and the dermatology and wound healing research communities at large, there are a number errors in the manuscript which must be addressed prior to further consideration for publication. 1. ROS is an abbreviation which stands for ‘reactive oxygen species’ however, on lines 41-42 the authors describe it as ‘radical oxygen species’. Indeed both the references used be the authors also correctly use the term reactive oxygen species. Please correct this error. : We have corrected it as reactive oxygen species in L42. 2. The statement “Prasiola japonica ethanol extract (Pj-EE), is a loliolide-rich source that has also been shown to possess antioxidant activity.” On lines 77-78 requires a reference. : We have included the reference [27] 27. Park S.H.; Choi E.; Kim S.; Kim D.S.; Kim J.H.; Chang S.; Choi J.S.; Park K.J.; Roh K.B.; Lee J.; Yoo B.C.; Cho J.Y. Oxidative Stress-Protective and Anti-Melanogenic Effects of Loliolide and Ethanol Extract from Fresh Water Green Algae, Prasiola japonica. Int J Mol Sci 2018, 19. 3. TH17 is written differently in the two instances on line 66. : We have fixed it as Th17 cell-produced cytokines (see L67-68) 4. Many of the figures have too many panels included, which becomes a problem where individual panels are then made too small to be clearly visualised. In particular, the H2DCF-DA staining in figure 2 is much smaller than the DAPI staining (figure 2d) and as such is very difficult to see and make out at all. Consider separating into multiple figures if appropriate, but at least ensure that the information can be clearly seen. : Due to connection between data in each figures, we would like to keep current panels in each figure. Instead, we have increased the size of Fig, 2c to discriminate each group. 5. Moreover, there are issues with the labelling of the panels in figure 2. The legend says that (c) is the H2-DCFDA–stained HaCaT cells treated with UVB and loliolide. But this is in fact the quantification graph of ROS generation (assuming that this is made from the staining above which does not have any panel identification. Please include identification of all panels and include description of ‘ROS generation’ quantification in the legend (ie is it intensity of fluorescent signal of H2DCF-DA>) and ensure that the methods are updated also. : We have included this like “(c) ROS generation in H2DCFDA–stained HaCaT cells treated with UVB and loliolide was analyzed by confocal microscopy and quantified by calculation of H2DCF-DA intensity signal using ImageJ.” in L109-111 and “The florescence intensity of ROS generating cells in three different pictures was quantified using ImageJ and afterwards the data was summarized in a plot” in L293-294. 6. Figure legend 2 also states that statistical analysis was performed both comparing to the normal (non-UV treated) and control (UV treated with no loliolide treatment). In that case please include ## in all panels where there are significant differences between the groups compared with the normal group. For example, in 2b where UV treatment clearly reduces cell viability, and likely is still significantly down in the loliolide 25 and 50 groups at least, ## should be included in line with the figure legends description. Stats (or at least significance markers) appear to be entirely missing from the graph in 2c. : We have significance marker in Fig. 2b as well as Fig. 2c bottom panel, according to this comment. 7. The results 2.2 on line 115 mention analysis of gene expression in H2O2-treated HaCaT cells. While this is also mention in the methods on line 300, no justification for using H202 rather than UVB irradiation is given, and significantly, the figure legends and all graphs state that the experiments were carried out in UV treated HaCaTs with or without loliolide treatment. This must be corrected. : We have fixed wrong adding of H2O2 to “UVB-irradiated HaCaT cells” in L122 and corrected like “Then, cells were pretreated with loliolide (0–100 µM) or Pj-EE (0–200 µg/mL). The cells were then exposed by UVB (30 mJ/cm2) and further incubated for 24 hours” in L333-334. 8. This sentence should also be reworded to “expression of MMP-1, MMP-9 (Figure 3a), MMP-2 (Figure 3b) and MMP-3 (Figure 3c) as well as restoring Sirt-1 expression (Fig. 3a)”. : We have reworded like “the expression of MMP-1 (Figure 3a), MMP-9 (Figure 3a), MMP-2 (Figure 3b), and MMP-3 (Figures 3c) as well as restoring Sirt-1 expression (Figure 3a” according to this comment (see L121-122). 9. Please also include stats details in the legend for figure 3. : We have included statistical evaluation details like “. Statistical significance was evaluated using the Kruskal–Wallis/Mann–Whitney U test. ## p < 0.05 compared with the normal group (b and c: UV - / Loliolide -) and ** p < 0.01 compared with the control group (b and c: UV + / Loliolide -)” in L130-132. 10. In the results 2.3 on line 125-126 is states that loliolide showed an increase in cell migration and wound repair in HaCaT cells in a dose-dependent manner (Figure 4a and 4b). However, the figures only present data for a single dose of loliolide (100u M) compared to control. This statement is therefore incorrect and should be changed. A similar statement is made in results 2.4 on line 153 referring to Pj-EE which was only assessed at 200ug/ml for the scratch wound assay : We have corrected wrong explanation like “loliolide (100 mM) showed an increase in cell migration and wound scratch closure in HaCaT cells (Figures 4a and 4b)” in L136-137 and like “Pj-EE (200 mg/mL) showed an increase in cell migration in HaCaT cells (Figure 5b)” in L166-167. 11. Both Figures (4b and 5c) need error bars and stats included and the images in panels 4a and 5b are very unclear and need to be improved so that the cells can be better visualised. : We have included error bars after calculating three experimental data and also tried to improve the quality of Fig, 4a and 5b. 12. The results 2.3 and 2.4 also talk about MMP-9 (lines 133 and 158), but the figures (4d and 5e) are labelled MMP-1. Both MMP-1 and MMP-9 primers are listed in table 1 of the methods and were analysed in figure 3a so I do not know what gene was analysed in this section. Also, the primers for KGF are missing. Please correct. : We have corrected it to MMP-1 in L144 and L171, and added KGF primer sequences in Table 1. 13. Both of these results section also incorrectly state that loliolide and Pj-EE showed a remarkable decrease in IL-1, IL-17 and IL-22 (lines 133-134 and 158-159) where the figures show that the expression of these markers are clearly increased. The discussion (lines 209-211) correctly states that the expression of these wound healing-promoting cytokines is enhanced. Please correct the results. : We have corrected this sentences according to this comment like “it showed a significant downregulation in the expression of inflammation-related genes MMP-1 and TNF-a (Figure 4d), while IL-1, IL-17, and IL-22 genes were dose-dependently increased (Figures 4e, 4f, and 4g)” in L143-145. 14. I would also suggest changing ‘cell migration and wound repair’ on line 125 to the more correct terms ‘cell migration and scratch wound closure’ and also ensuring that the order of the genes presented in the results (eg line 133 MMP-9, TNFa, COX2) matches the order of the genes in the figures (figure 4d TNFa, MMP-1, COX-2) throughout. : We have changed them to “wound scratch closure” and “TNF-a, MMP-1, and interleukins”, according to this comment in L136 and L141, 15. The legend of Figure 4 needs to include ‘and inflammatory genes’ after migration on line 142 to match the data presented and also the legend of figure 5. : According to this comment, we have added the keywords in L151, 153, 176, and 179. 16. Figure 6 is not very useful. Without inclusion of arrows to describe the effect of loliolide and Pj-EE on the genes listed (eg up regulation or down regulation) the flow chart is meaning-less. Perhaps remove this figure entirely and spread out the panels from other figures for easier reading? : This is good point. Since each panel in each figure is correlative, we would like to keep current form. Also, because we have handled a lot of molecular and cellular parameters in this study, we would like to also keep scematic summary figure to help reader’s understanding, according to journal’s guidline. 17. Lines 185-187 makes a strange sentence and should be reworded. Also, including mention of figures in the discussion seems strange considering this is already presented in the results. : We have rephrased to improve its meaning like “In addition to presenting a good antiapoptotic effect, loliolide also dose-dependently reduced not only oxidative stress (Figure 2d), but also DNA damage (Figure 2e), cell death (Figure 2f), and apoptosis-related caspase activation (Figure 2g) exhibited under UV irradiation conditions” in L202 to 204. 18. Please update the methods (line 233-235) to indicate that these are antibodies, list the catalogue number and also include details of the secondary antibodies and detected methods used. : We have included all details about these in L252-258 like “Primary antibodies to phospho-, total or cleaved forms of PI3K (PI3K; CST # 4292, and p-PI3K; CST #4228), AKT (AKT; CST #9272, and p-AKT; CST # 4058), ERK (ERK; CST # 4696, and p-ERK; CST # 9101), caspase 3 (procaspase3; CST # 9665, and cleaved caspase3; CST # 9664), caspase 8 (procaspase8; CST # 4790, and cleaved caspase8; CST # 9496), caspase 9 (procaspase9; CST # 12827, and cleaved caspase9; CST # 7237), and β-actin (CST # 4967), and horse radish peroxidase (HRP)-labeled secondary antibodies to anti-mouse and anti-rabbit IgG were obtained from Cell Signaling Technology (Danvers, MA, USA).”. 19. The methods 4.3 states that or determining the compound’s cytotoxicity, the same method was applied using only PJ-EE (0–100 μg/mL) for 24 hours. This appears incorrect as the loliolide was investigated in normal HaCaTs at 0-100uM in figure 2a and Pj-EE at 0-400ug/ml in figure 5a. Please correct. : We have fixed these like “loliolide (0–100 µM) and Pj-EE (0–400 µg/mL)” in L276. 20. Methods 4.6 also needs to be expanded to include the treatment time prior to FACS analysis and more details about the order of irradiation and treatment. All the previous experiments were performed with pretreatment followed by irradiation, the methods state loliolide treatment after UVB irradiation (line 277) and the results say is was simultaneous (line 101). Please clarify. : We have clarified this like “Cells were pretreated with or without loliolide (0–100 µM) and then further irradiated with UVB (30 mJ/cm2)” in L307-308 and “When the UVB-treated cells were pretreated with loliolide” in L104-105. 22. Please included the quantity of template used in the RT-PCR analysis (methods 4.8) and also clarify why two different methods for analysing gene expression (gels versus fold change) were used. : We used 4 mM of 5’ and 3’ primers per reaction. Each primer for RT-PCR or real-time PCR was selected since it was showing best conditions. To have best data, we could do both methods according to different primer. 23. Methods 4.9 ends by stating that debris was removed by PBS. Please expand and included details about whether normal culture media or treated media was replaced in the wells. : We have included some sentences to explain more like “Scratch wounds were created mechanically with a sterile pipette tip (Ø = 0.1 mm), whose size range varied from approximately 0.5 mm to 0.9 mm in width. Detached cells and debris were washed away with PBS solution and fresh medium was treated to the wells” in L343-345. 24. Finally, what is the concentration of loliolide compound within the Pj-EE extracts at the concentrations presented? Is it likely that the 200ug/ml Pj-EE dose is delivering an equivalent amount of loliolide at the effective dose of 100uM or could it be higher? This information should be included. : We could not determine the exact concentration of loliolide in Pj-EE. We have just followed previous report on this algea. Since we totally agree with this comment, we will further examine the level of loliolide from Pj-EE using GC. 25. Also, why were the UV-protective effects of Pj-EE not investigated? The methods 4.3 state that UVB radiation was performed and Pj-EE treatment at 0-200 ug/ml performed but I can’t see this in the results. Only scratch data is presented in figure 5. : There was wrong, so we have fixed this in L276-279 to make it clear like “HaCaT cells were seeded at a density of 4 × 104 cells per well in a 96-well plate for 24 hours and then treated with loliolide (0–100 µM) and Pj-EE (0–400 µg/mL) for 24 hours. For testing UV-protective activity, HaCaT cells were pretreated with loliolide (0–100 µM) and then further exposed to UVB radiation (30 mJ/cm2). The viability of HaCaT cells treated with loliolide or Pj-EE or UV-irradiated HaCaT cells pretreated with loliolide was measured using the MTT assay” in L276-279 of p13. 26. Can this also be performed and included in the manuscript? : We have data but we are currently preparing other paper to present the anti-oxidative and photoaging activity of Pj-EE. |
Round 2
Reviewer 1 Report
The title is unsual. I would replace antiscratching with "wound healing"
Author Response
We have changed the tittle like “Loliolide presents antiapoptosis and wound healing effects in human keratinocytes “ in L2 of p1.
Reviewer 2 Report
The authors did not understand my previous comments - they used a "scratch" assay to examine cellular migration which is not the same as "scratching". Scratching is a human response to itch - using fingernails on itching areas. The title should not state "antiscratching" but improved cellular migration. They need to change that whole section. Otherwise it is OK.
Author Response
The authors did not understand my previous comments - they used a "scratch" assay to examine cellular migration which is not the same as "scratching". Scratching is a human response to itch - using fingernails on itching areas. The title should not state "antiscratching" but improved cellular migration. They need to change that whole section. Otherwise it is OK.
: Since reviewer No.1 also recommended replace of scratching to wound healing, therefore, we have fixed them to wound or wound healing (thus, antiscratching -> wound healing , Scratching-> wound) (please see L32, L33, L35 L82 L132 L149, L158, L159, L163 L174 L222, L229, and L339).
Reviewer 3 Report
All the comments were addressed and sufficiently considered and included in the new version of the manuscript. Therefore, the manuscript has been significantly improved.
Author Response
All the comments were addressed and sufficiently considered and included in the new version of the manuscript. Therefore, the manuscript has been significantly improved.
: Thanks very much.